# ToxDAR: A Workflow Software for Analyzing Toxicologically Relevant Proteomic and Transcriptomic Data, from Data Preparation to Toxicological Mechanism Elucidation

**DOI:** 10.3390/ijms25179544

**Published:** 2024-09-02

**Authors:** Peng Jiang, Zuzhen Zhang, Qing Yu, Ze Wang, Lihong Diao, Dong Li

**Affiliations:** 1School of Basic Medical Sciences, Anhui Medical University, Hefei 230032, China; tmcjiangp@gmail.com (P.J.); zzz.13325@163.com (Z.Z.); 2College of Life Sciences, Hebei University, Baoding 071002, China; yuuqing2021@163.com; 3State Key Laboratory of Medical Proteomics, Beijing Proteome Research Center, National Center for Protein Sciences (Beijing), Beijing Institute of Lifeomics, Beijing 102206, China; wangze_bprc@163.com (Z.W.); lhdiao@126.com (L.D.)

**Keywords:** toxicological transcriptomics analysis, toxicological mechanisms, R package

## Abstract

Exploration of toxicological mechanisms is imperative for the assessment of potential adverse reactions to chemicals and pharmaceutical agents, the engineering of safer compounds, and the preservation of public health. It forms the foundation of drug development and disease treatment. High-throughput proteomics and transcriptomics can accurately capture the body’s response to toxins and have become key tools for revealing complex toxicological mechanisms. Recently, a vast amount of omics data related to toxicological mechanisms have been accumulated. However, analyzing and utilizing these data remains a major challenge for researchers, especially as there is a lack of a knowledge-based analysis system to identify relevant biological pathways associated with toxicity from the data and to establish connections between omics data and existing toxicological knowledge. To address this, we have developed ToxDAR, a workflow-oriented R package for preprocessing and analyzing toxicological multi-omics data. ToxDAR integrates packages like NormExpression, DESeq2, and igraph, and utilizes R functions such as prcomp and phyper. It supports data preparation, quality control, differential expression analysis, functional analysis, and network analysis. ToxDAR’s architecture also includes a knowledge graph with five major categories of mechanism-related biological entities and details fifteen types of interactions among them, providing comprehensive knowledge annotation for omics data analysis results. As a case study, we used ToxDAR to analyze a transcriptomic dataset on the toxicology of triphenyl phosphate (TPP). The results indicate that TPP may impair thyroid function by activating thyroid hormone receptor β (THRB), impacting pathways related to programmed cell death and inflammation. As a workflow-oriented data analysis tool, ToxDAR is expected to be crucial for understanding toxic mechanisms from omics data, discovering new therapeutic targets, and evaluating chemical safety.

## 1. Introduction

Toxicological mechanism research is crucial for revealing the impact of chemical substances at the molecular, cellular, and even organ levels on living systems [1,2]. It is essential for predicting and assessing chemical risks [3], establishing chemical safety standards, and developing safe chemical substitutes [4,5]. Moreover, it plays a key role in determining clinical treatment dosages, reducing the risk of side effects [6], and advancing personalized medicine [7,8].

In recent years, transcriptomics and proteomics have increasingly become essential tools for studying the interactions between toxins and organisms [9]. While genomics provides the genetic blueprint of an organism [10], its relatively static data do not capture an organism’s rapid response to environmental changes. Transcriptomics allows for the holistic study of RNA expression changes [11], offering direct evidence of how genes regulate in response to external toxins. Concurrently, proteomics, by examining the expression, modification, and interaction of proteins, reveals their specific alterations under the influence of toxins, thus elucidating the mechanisms of toxicology directly [12,13,14]. Although the advancement of ‘omics’ technologies has greatly enhanced our capacity to study the effects of toxin exposure [15], the vast datasets pose significant challenges for bioinformatics [16]. Current toxicological databases such as LINCS, ToxCast [17], and Open TG-GATEs [18] and analytical tools like TCGAbiolinks2.32.0 [19], RTCGAToolbox2.34.0 [20], and cBioPortalv5 [21] are available. Yet, they only facilitate basic analysis processes like data download, preprocessing, and comparative analysis, which do not meet the needs of toxicological mechanism research. Other general analysis software, such as PCA1.0.7 [22], limma [23], clusterProfiler4.12.6 [24], and Cytoscape3.10.2 [25], can perform targeted data analysis functions, such as differential expression, functional enrichment, and network construction. Additionally, there have been some efforts in building toxicology-related knowledge bases. For example, the National Center for Toxicogenomics has developed the Chemical Effects in Biological Systems (CEBS) knowledge base [26], which includes toxicology-related literature and dataset information. The Comparative Toxicogenomics Database (CTD) [27] integrates toxicology information on chemicals, genes, phenotypes, and exposures, revealing the impact of environmental factors on disease etiology and molecular mechanisms. The ECOTOXicology Knowledgebase [28] aggregates various ecological toxicity data and supports risk assessment for ecological toxicity testing. However, the field of toxicology still lacks a comprehensive knowledge base that systematically presents the relationships between toxic molecules and biological systems, as well as omics data analysis software based on such a knowledge base. More importantly, researchers must frequently switch between different analytical software tools and online resources during the research process, which can be complex, time-consuming, and prone to errors. Therefore, it is necessary to develop a procedural system for the analysis of toxicogenomic data that improves the accuracy, reliability, and reproducibility of data analysis.

Addressing the challenges in the field of toxicogenomics data (including Toxicoproteomic and Toxicotranscriptomic data) analysis, this article presents an innovative solution, ToxDAR, designed to enhance the capabilities of data analysis and annotation, thereby improving the efficiency and quality of toxicological mechanism elucidation. Implemented as an R package, this solution integrates a wide range of analysis functions commonly used in the analysis of toxicogenomic data, including data reading, preprocessing, differential analysis, functional annotation, and network analysis. The ToxDAR package also incorporates annotation information from multiple databases and knowledge bases, encompassing associations between toxins and diseases, pathways, genes, and other entities, thus providing a knowledge framework for the analysis of toxicogenomics data. ToxDAR offers a suite of tools for analyzing and interpreting toxicogenomics data, promising to provide deeper insights into the mechanisms of toxicology.

## 2. Result

### 2.1. Software Framework

The ToxDAR software package is a bioinformatics analysis system tailored for the analysis of toxicogenomics data to identify relevant biological pathways associated with toxicity from the data and to establish connections between omics data and existing toxicological knowledge. It comprises four key modules (Figure 1): (I) Data Preparation, (II) Quality Control, (III) Data Analysis, and (IV) Data Interpretation. In the Data Preparation, Quality Control, and Data Analysis modules, ToxDAR integrates software packages such as NormExpression, DESeq2, igraph, as well as functions like prcomp and phyper to enable various analysis functionalities for toxicogenomics data, including data preparation, quality control, differential expression analysis, functional analysis, and network analysis. In the Data Interpretation Module, the software package provides knowledge annotation and validation for the analysis results by leveraging underlying toxicological domain-specific knowledge. ToxDAR integrates knowledge spanning five major categories of entities—toxins, genes, biological pathways, diseases, and phenotypes—and delineates 15 types of relationships among these entities, offering detailed knowledge support for the analysis of toxicological mechanisms. ToxDAR provides various forms of annotation for analysis results. For instance, it annotates differential analysis results using compound-gene relationships gathered from the Comparative Toxicogenomics Database (CTD), facilitating rapid screening and the identification of candidate molecules. It also utilizes harmful outcome pathways associated with compounds from the Adverse Outcome Pathway (AOP) wiki to validate the reliability of specific toxicogenomics data analysis results. By consolidating multiple analysis functions and knowledge resources, ToxDAR establishes a streamlined analysis workflow, simplifying the processing of large and complex toxicogenomics datasets.

### 2.2. Software Function

The Data Preparation Module plays a pivotal role in addressing batch effects [29] in toxicogenomics data. In toxicogenomics experiments, batch effects often arise due to variations in laboratory conditions, reagent batches, and personnel, which can affect the experimental data. To mitigate these potential biases and noise, normalization procedures are essential [30]. However, selecting the optimal normalization method for a given dataset is a challenging task [31], often complicated by the unknown origins of these biases. The ToxDAR platform integrates ten different normalization techniques, including the median of the ratios of observed counts (DESeq) [32,33], upper quartile (UQ) [34,35], Trimmed Mean of M values (TMM) [36,37,38], Total Ubiquitous (TU), Total Read Count (TC), Total Read Number (TN) [39], External RNA Control Consortium (ERCC) [40,41], Housekeeping Genes (HG7) [42], Cellular RNA (CR) [43], and Nuclear RNA (NR) [44]. More detailed information about these techniques is provided in the Appendix A. Through ToxDAR, researchers can obtain comprehensive analytical reports, enabling them to compare and select the normalization scheme best suited to their research objectives. The evaluation of normalization results is based on the AUCVC (Area Under the normalized Coefficient of Variation threshold Curve) and the mSCC (Median Spearman’s Rank Correlation Coefficient) metrics. The AUCVC metric represents the variation in the number of uniform genes (defined as genes with standardized expression levels with a Coefficient of Variation (CV) below a pre-set threshold across all samples) as the normalization CV threshold varies. When the normalized expression levels of uniform genes have sufficiently low CVs, it indicates higher consistency across different samples, implying that technical noise has been more effectively reduced. The mSCC metric reflects the proportion of gene pairs with corresponding Spearman correlation coefficients across the entire dataset. By calculating the Spearman correlation coefficients between each pair of genes (normalized vs original data), we obtain the median of all gene pairs’ Spearman correlation coefficients, which is used to assess the efficacy of the data normalization scheme. In the context of gene expression analysis, an mSCC value approaching zero indicates that the normalized data exhibit a lower correlation with the raw data at the level of gene expression.

The Quality Control Module is designed to test the reliability of data, filtering out those suitable for further analysis. Utilizing Principal Component Analysis (PCA), ToxDAR can reduce the dimensionality of the vast number of gene variables disturbed after toxic exposure and extract their main characteristics. By assessing data variance, it visualizes the variations of high-dimensional data in a low-dimensional space. The PCA plot intuitively displays the variability between different experimental groups and individual samples within groups, revealing differences in data distribution. The clear spatial differences between different experimental groups indicate better data quality. Moreover, ToxDAR allows for the separate visualization of gene expression data distributions for different experimental groups. Through violin plots, it intuitively compares the data distribution differences between groups, considering the dimensions of all gene expressions. The data distributions across different experimental groups show uniformity, indicating higher data quality.

The Differential Analysis Module of the ToxDAR software package, based on the DESeq2 framework, efficiently processes time-series-like data generated from toxicology experiments. This module employs a statistical model grounded in the negative binomial distribution to identify differentially expressed genes [45], estimating the probability of gene expression differences between samples through a negative binomial generalized linear model [46]. Through this process, it thoroughly accounts for the discreteness and variability of gene expression data, as well as the impact of library size differences on differential analysis, thereby enhancing the accuracy of the results. ToxDAR automates the integration and formatting of multiple comparative datasets, revealing commonalities in toxicological effects or intra-group expression variability, obviating the need for manual data handling and restructuring. Users can, depending on their research objectives, choose to combine differentially expressed genes identified under various conditions using either a union or intersection approach. Moreover, ToxDAR identifies known genes associated with the toxin, based on the toxin ID (MeSH ID) provided by the user from its built-in knowledge database, and then performs intersection analysis with the list of differentially expressed genes to pinpoint known targets of the toxin, providing relevant literature and association scores as supporting evidence. The software package offers a volcano plot visualization tool to graphically display changes in gene expression levels and annotate the names of known toxin-related genes alongside corresponding nodes, aiding researchers in rapidly identifying and interpreting genes with significant differential expression.

The Functional Annotation Module of ToxDAR provides users with hierarchical functional categorization information of genes, facilitating a systematic understanding of molecular functions and identifying molecular entities involved in multiple key biological pathways. ToxDAR conducts enrichment analyses on differentially expressed genes following toxin exposure using the hypergeometric distribution method [47], based on integrated annotation datasets such as GO [48], DO, and KEGG [49], describing the biological functions of differential genes across multiple dimensions. Details of the enrichment analyses methods and datasets are provided in the Appendix A. Utilizing the ssGSEA algorithm, ToxDAR processes the list of differentially expressed genes post-toxin exposure, translating the differences in gene expression into levels of biological pathway activation or suppression, thereby revealing the molecular mechanisms of the organism’s response to toxin exposure. Furthermore, this module employs various graphical presentation methods, such as bar charts for a visual reflection of the functional enrichment status of DEGs. Chord diagrams illustrate the interconnections between enriched functional entries and DEGs, as well as the functional correlations amongst different DEGs. Clustering diagrams are used to display the distribution patterns of DEGs in functional classifications. Enrichment curve graphs indicate the significance of different functional collections in gene expression data. Heatmaps are utilized to visually present the activation or suppression status of biological pathways after a toxin’s effect, providing researchers with a comprehensive and intuitive analytical perspective and aiding in the rapid understanding of related analysis results.

The Network Analysis Module encompasses functions such as the parsing of network topological structures, implementation of network clustering, and exploration of network associations. Leveraging its built-in protein–protein interaction database, the ToxDAR software is capable of efficiently constructing molecular interaction networks and utilizing its integrated network algorithms to carry out tasks such as network clustering analysis and node degree calculation. Systems biology approaches allow us to understand molecular interactions within biological systems from a holistic perspective and elucidate the specific impacts of toxins on these interactions. With network analysis techniques, we can represent these interactions in the form of networks, enabling in-depth studies on how toxins disrupt cellular signaling, metabolic pathways, and the functioning of gene regulatory networks. This comprehensive methodology enables researchers to identify key components and nodes within biological processes and to reveal the intrinsic links between toxins and critical genes, biological pathways, and phenotypes across multiple dimensions. It offers a systematic perspective to understand the toxicological mechanisms triggered by toxin exposure, aiding researchers to more comprehensively grasp the complexity of biological systems [50].

The Data Interpretation Module of the ToxDAR software package amalgamates an extensive range of toxicological knowledge, offering detailed annotations for five major entity categories: toxins, genes, biological pathways, diseases, and phenotypes, along with information on 15 types of relationships between them. This provides a solid knowledge base for the in-depth analysis of toxicological mechanisms. It includes interpretation strategies involving toxin-related genes and gene regulatory networks, applying multi-dimensional precise annotations to analysis results to facilitate the exploration of molecular mechanisms following toxin exposure. For instance, the ToxDAR software is capable of utilizing relationship data between compounds and genes provided by the CTD to rapidly filter results of differential expression and identify potential candidate molecules. It also employs information on compound-related adverse outcome pathways provided by the Adverse Outcome Pathway (AOP) Wiki to thoroughly annotate specific toxicogenomics data analysis results.

### 2.3. Research Case: Toxicological Mechanism Analysis of Public Transcriptome Data in L02 Cell Line Post-Triphenyl Phosphate (TPP) Exposure

This study employs transcriptomic data submitted by Xiaoqing Wang et al. and uses ToxDAR to explore the impact of triphenyl phosphate (TPP) on the L02 cell line at the omics level [51], unveiling and interpreting the potential toxicological mechanisms of TPP exposure. The dataset encompasses 20 expression profile data obtained after treating the L02 cell line with various concentrations of TPP by Xiaoqing Wang and colleagues. Initially, to eliminate technical biases and batch effects, we preprocess the raw data from Xiaoqing Wang et al.’s dataset using multiple normalization methods integrated into ToxDAR, followed by quality control of these omics data. Subsequently, employing the differential expression analysis module of ToxDAR, we identified differentially expressed genes associated with TPP exposure. Further, we annotated the function similarity and bias of differentially expressed genes at various concentrations and time points using the software’s annotation function and marked their association with TPP exposure. Around the key molecules identified, we utilized ToxDAR to construct molecular interaction networks, revealing the interconnections between toxins and key genes, biological pathways, phenotypes, and other dimensions, providing a systemic perspective for understanding the toxicological mechanisms related to TPP exposure.

Initially, we utilize the ToxDAR software package to normalize the transcriptome data post-TPP exposure using ten different methods, subsequently generating a report on the effects of normalization. These normalization methods are evaluated based on two metrics: AUCVC (Figure 2A) and mSCC (Figure 2B), to select the appropriate normalization method. For the transcriptome dataset following post-TPP exposure, it was discovered that the normalization using the Upper Quartile (UQ) method yielded the highest AUCVC value while also having an mSCC value closest to zero. Therefore, the UQ method can be considered the optimal option for data preparation in this dataset.

Subsequently, we conducted a quality control analysis of the data using the Data Quality Control Module of the ToxDAR software package. We generated PCA plots and violin plots of gene expression distribution for these datasets, allowing us to visually observe the variability both between different experimental groups and within the same group across samples. The PCA results (Figure 3A) reveal significant differences in the spatial distribution among experimental groups, while the violin plot results (Figure 3B), considering all dimensions of gene expression, display uniformity of data distribution across experimental groups. This indicates a high quality of data, suggesting that these datasets are suitable for further data mining and analysis.

Afterward, we utilized the differential analysis module of the ToxDAR software package to identify differentially expressed genes associated with TPP exposure at various concentrations and time points (Figure 4). Additionally, the integrated knowledge base within ToxDAR provided clear associations between differentially expressed genes and TPP exposure. Among the identified differentially expressed genes, ABCC3, MYC [52], and STAC3 have been previously confirmed by research to be associated with TPP exposure.

To gain a comprehensive understanding of the functions of these genes, we proceeded to perform enrichment analysis on the differentially expressed genes following TPP exposure using multiple annotation datasets including GO, DO, KEGG, and others collected within the software package (Figure 5A). This analysis aimed to elucidate the biological roles of these differential genes from multiple perspectives, including function and disease. The results of the analysis indicated that TPP exposure is associated with signaling pathways related to cellular programmed death, inflammatory responses, and others, as depicted in the chord diagram (Figure 5B). These key pathways share several pivotal molecules, such as ABCC3 and THRB. This provides important scientific evidence for further exploration of the toxicological mechanisms of TPP (Figure 5C).

This study conducted an in-depth analysis of the differentially expressed gene lists under varying exposure concentrations using the ssGSEA function integrated within the software package (Figure 6A). The ssGSEA method quantifies changes in gene expression profiles as a result of toxic exposure into the activation or inhibition states of biological pathways, revealing the mechanisms by which the organism responds to exposure at a molecular level. The changes are presented visually through enrichment plots and heatmaps for intuitive representation (Figure 6B). The results of the study showed that in samples exposed to a TPP concentration of 881 mg/kg, there was a notable activation of the biological pathway for cell apoptosis. Additionally, in the same samples exposed to 881 mg/kg TPP, there was a significant inhibition of the phenylalanine metabolism pathway. Furthermore, as the exposure concentration of TPP increased, there was a trend of enhanced activation of signaling pathways related to inflammatory responses.

In conclusion, based on the analysis of significant molecules related to the toxicological mechanisms of TPP, such as THRB, ABCC3, and NOTCH1, we employed the ToxDAR network analysis and annotation module to map the associations between the toxin and key molecules, biological pathways, and phenotypes (Figure 7A). Within this network, TPP is linked with biological pathways such as apoptosis and inflammatory responses, indicating that triphenyl phosphate (TPP) may act as a potential endocrine disruptor. It exerts its effects by activating the thyroid hormone receptor β (THRB) molecule, thereby influencing signaling pathways related to apoptosis and inflammatory responses, ultimately adversely affecting thyroid function. This is consistent with the knowledge of adverse reaction pathways caused by TPP compounds documented in the literature. It provides a systematic perspective and in-depth insights into the toxicological mechanisms associated with TPP exposure, further confirming the accuracy and reliability of our analysis (Figure 7B). We have added the flowcharts to GitHub (https://github.com/TMCjp/ToxDAR, accessed on 29 August 2024).

In summary, the ToxDAR software package supports the research into the toxicological mechanism of TPP on two levels: Firstly, through its omics data analysis capabilities, ToxDAR enables standard preprocessing and effective quality control, providing a list of differential genes post-exposure to toxins along with corresponding functional annotations, thereby establishing a rapid data analysis workflow. Secondly, with the integration of databases and algorithms, ToxDAR offers in-depth mechanistic insights, allowing us to understand toxicological effects at a systems biology level. The comprehensive functions of ToxDAR serve as an effective tool for delving into the key toxicological mechanisms of TPP and charting its toxicity profile, which is crucial for future risk assessments and toxicological research.

## 3. Discussion

The analysis and interpretation of toxicogenomics data play a pivotal role in biological research. Despite the availability of various tools to execute specific steps in the analysis, previous studies often lacked a comprehensive and user-friendly tool. ToxDAR, as an R software package, provides a convenient and efficient tool for the analysis of high-dimensional data generated through omics technologies. It serves as a critical resource for the analysis and interpretation of such data, enhancing our understanding of complex biological processes and offering deeper insights into the toxicological mechanisms of specific chemicals. The package integrates multiple omics data analysis functions, providing easy installation and invocation methods, and enabling data preparation, differential expression analysis, functional annotation, and network analysis of toxicogenomics data. Additionally, the package incorporates various annotation resources, including pathway information, ontology terms, and the relationships between toxins and multiple entities, offering rich contextual information for the interpretation of toxicogenomics data.

Our software can be applied to various issues in the field of toxicology research, such as dose–effect prediction and the study of new alternatives for toxicity testing. In dose–effect prediction, our software can examine the impact of different doses of toxins on the organism and molecular changes to determine the effect of dosage on toxicological mechanisms. This provides possibilities and support for further establishing dose–response relationships in toxicology. In the case study of triphenyl phosphate (TPP) that we provide, we explore the changes in differentially expressed genes and the effects on biological pathway activities in the L02 cell line at various concentrations of TPP. In terms of new alternatives for toxicity programs, because traditional animal models may not accurately reflect the actual conditions of clinical patients and exhibit species differences, the academic community has gradually developed new systems such as in vitro cell models like organoids. These systems include liver, kidney, and lung organ-on-chip models and multi-organ-on-chip models designed for toxicological assessment [53,54]. Our software can evaluate the accuracy and stability of these organ-on-chip models by examining the differences in toxin-responsive protein expression and pathway activities between organ-on-chip models and traditional animal models.

Leveraging ToxDAR’s flexible framework and cross-system technical architecture, we plan to aggregate more comprehensive methods and data in the future, effectively integrating them into a common framework for researchers to extract meaningful information, thereby more profoundly characterizing and understanding the toxicological mechanisms within the data.

## 4. Materials and Methods

### 4.1. Primary Functions

ToxDAR is developed using the R programming language and integrates numerous commonly used packages provided by Bioconductor. ToxDAR utilizes the NormExpression package for the processing and normalization of toxicogenomics expression profile data, thus eliminating potential biases and noise within the dataset. It uses the built-in prcomp function in R for principal component analysis (PCA) of the preprocessed toxicogenomics data, reducing the dimensionality of variables and extracting key features to segregate samples and identify outliers, thereby assessing the quality of the data. Differentially expressed genes are generated using the widely employed DESeq2 package [55]. The phyper function is utilized to conduct hypergeometric tests. The GSEA package [56] is employed to translate changes in gene expression levels post-toxin exposure into the activation or inhibition states of biological pathways. Complex network analysis is performed using the igraph package [57], which includes network topology analysis, clustering, and the visualization of various layouts. Additionally, ToxDAR integrates the dplyr, ggplot2 [58], and Complex Heatmap packages [59] to facilitate data manipulation and visualization functionalities [60] (Table 1).

### 4.2. Data Sources

The ToxDAR software package extracts associations and supporting evidence between toxicants and biological entities such as genes, pathways, and phenotypes from multiple knowledge bases (Table 2). This includes the relationships and evidence between toxicants and genes, toxicants and pathways, and toxicants and diseases from the CTD database [61]. It also comprises the adverse outcome pathways knowledge from the AOP-Wiki database [62], which reflects toxicodynamics processes including key molecules, critical pathways, and the adverse outcomes they cause.

The ToxDAR software package concurrently compiles information from multiple databases encompassing biological pathways, diseases, and protein interactions. It aggregates associations between biological pathways and genes from the KEGG database [49] and integrates these with the correlations between toxins and pathways gathered from the CTD. Furthermore, the software collects data from the STRING database [63] on protein–protein interactions, as well as reliable associations between diseases and genes from DisGeNET [64].

Given the discrepancies in the representation of the aforementioned knowledge across various databases, this study employs the use of Disease Ontology (DO) terms [65] to match disease entities, Human Phenotype Ontology (HPO) terms [66] for phenotype entity matching, and Kyoto Encyclopedia of Genes and Genomes (KEGG) ontology terms for pathway entity alignment, thereby achieving cross multiple source integration.

### 4.3. Toxicological Classification System

To elucidate the mechanisms of toxicology within toxicogenomics data, the classification of toxins in this article is informed by the “Principles of Forensic Toxicology”, 5th edition by Barry S. Levine [67]. A toxin classification system has been constructed based on toxicological action mechanisms (Appendix A). Within each category, representative information about toxins has been manually augmented through a comprehensive review of knowledge databases and literature searches (Appendix A).

## Figures and Tables

**Figure 1 ijms-25-09544-f001:**
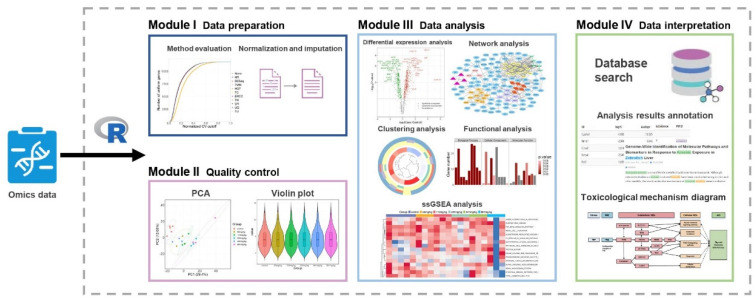
Framework of the ToxDAR Software Package. ToxDAR is written in R and is designed to handle omics quantitative expression profile data following exposure to toxins. ToxDAR contains four modules: Data preparation, Quality control, Data analysis, and Data interpretation. (I) Within the Data preparation module, the package integrates ten standardization methods and automatically evaluates the most suitable normalization approach for specific datasets. (II) In the Quality control module, principal component analysis and data distribution visualization are employed to assess the quality of the omics data. (III) In the Data analysis module, ToxDAR implements analytical functions such as differential analysis, functional analysis, and network analysis, and provides corresponding visualization schemes. (IV) The Data interpretation module utilizes domain-specific prior knowledge collected by ToxDAR. It not only annotates the results of omics analysis but also integrates the contextual information to elucidate the toxicological mechanisms of toxins.

**Figure 2 ijms-25-09544-f002:**
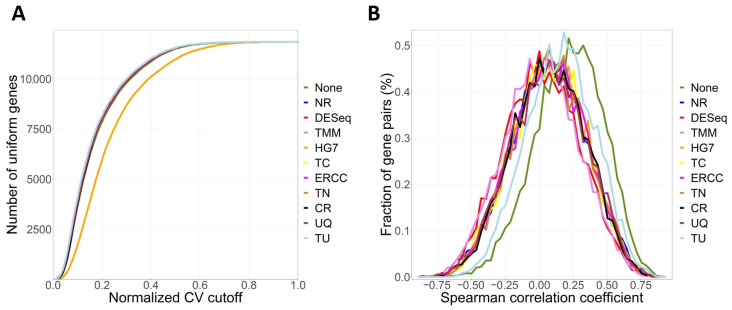
Assessing the normalization techniques for preprocessing of diverse quantitative datasets. (**A**) Index of AUCVC. (**B**) Metric of mSCC. The curves depicted in various colors correspond to distinct normalization methods. The label “None” denotes the absence of normalization. “NR” stands for the Nuclear RNA approach, “DESeq” signifies the method based on the median of ratios of observed counts, “TMM” refers to the Trimmed Mean of M-values method, “HG7” represents the Housekeeping Genes approach, “TC” indicates the Total Read Count method, “ERCC” is the method of the External RNA Control Consortium, “TN” symbolizes the Total Read Number approach, “CR” pertains to the Cellular RNA method, “UQ” denotes the Upper Quartile method, and “TU” stands for the Total Ubiquitous method.

**Figure 3 ijms-25-09544-f003:**
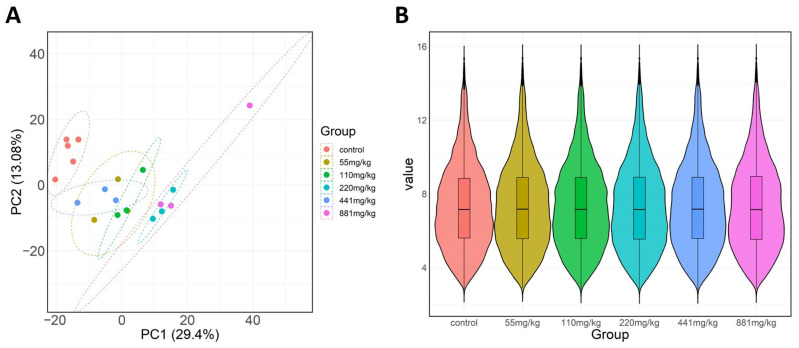
Quality Control Analysis of the Transcriptomic Data Set Following TPP Exposure. (**A**) PCA Plot: This plot, distinguished by points of varying colors representing different toxic exposure concentrations, clearly reveals the variability both between and within experimental groups. It demonstrates significant differences in the spatial distribution among the groups. (**B**) Violin Plot: This plot, differentiated by colors representing different toxic exposure concentrations, displays the variability in data distribution across different experimental groups.

**Figure 4 ijms-25-09544-f004:**
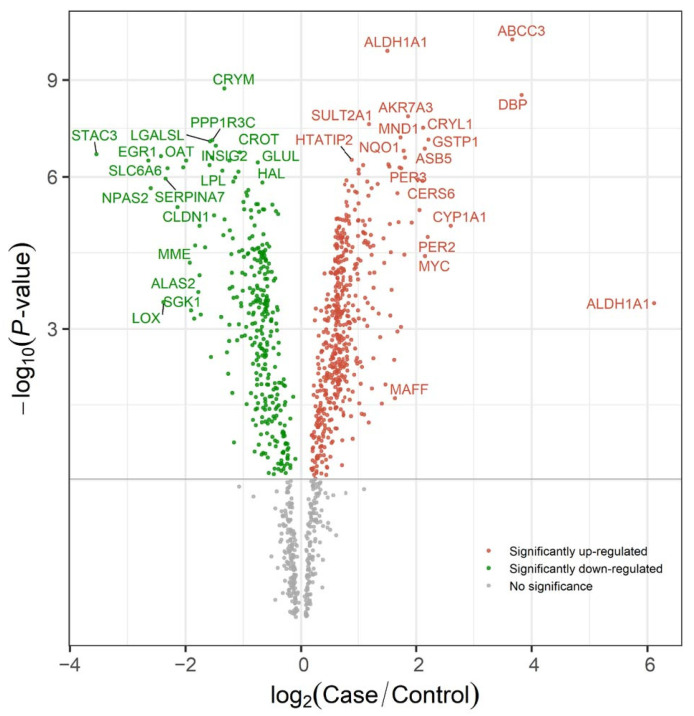
Volcano plot revealing differentially expressed genes under TPP exposure compared to the blank control group. The horizontal axis of the plot represents the log2 ratio of fold change in gene expression (Case/Control) and the vertical axis represents the −log10 (*p*-value), indicating the significance of the difference in gene expression. Red nodes in the plot represent genes that are significantly upregulated in the experimental group relative to the control group, while green nodes indicate genes that are significantly downregulated. Nodes labeled with gene names are those that have been confirmed to be related to the toxicological mechanism of TPP in the knowledgebase integrated within the ToxDAR software package.

**Figure 5 ijms-25-09544-f005:**
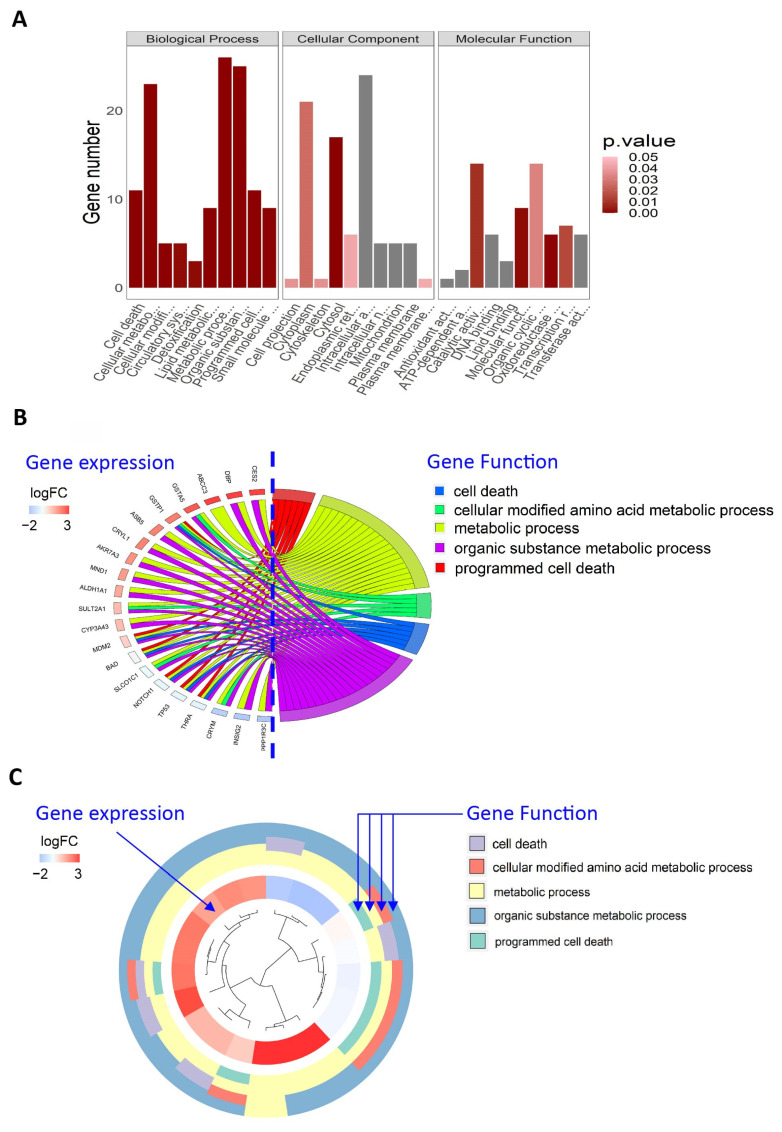
The GO functional enrichment analysis of differentially expressed genes after TPP exposure. (**A**) The diagram elaborates on the enriched functionalities of the differential genes based on the three main categories of GO: Biological Process (BP), Cellular Component (CC), and Molecular Function (MF). (**B**) A chord diagram reveals the associations between enriched functional entries and differential genes, also showing the functional correlations among different differential genes. The left side of the graph lists the differential genes, with node colors representing the logarithmic (log) values of differential multiples. The right side displays the enriched functional entries, differentiated by node colors. (**C**) A clustering diagram shows the functional clustering of the differential gene sets in the GO terms. The hclust method is used for hierarchical clustering of the differential gene expression profiles. The dendrogram next to it has its first ring representing the log fold change (logFC) of the genes—essentially the leaves of the clustering tree. Each subsequent ring outside represents the functional entries assigned to the genes.

**Figure 6 ijms-25-09544-f006:**
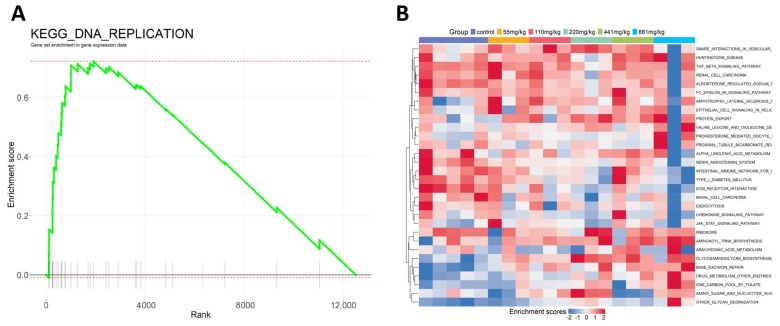
The impact of various concentrations of TPP on the biological pathway activities in L02 cells. (**A**) Gene set enrichment analysis of differentially expressed genes under the exposure condition (881 mg/kg) compared to a blank control. Enrichment curves reveal the enrichment of different functional sets in gene expression data. The x-axis label shows the cumulative ranking of the gene sets, while the y-axis label shows the enrichment score. (**B**) ssGSEA analysis of the differentially expressed genes at different exposure concentrations. The horizontal axis represents individual samples at varying exposure concentrations with bar graphs in different colors categorizing the samples by exposure levels (from left to right: Control, 55 mg/kg, 110 mg/kg, 220 mg/kg, 441 mg/kg, 881 mg/kg). The vertical axis represents various biological pathways. The colors of the samples on the heatmap correspond to different exposure concentrations, and the heatmap colors indicate the degree of activation or inhibition of the biological pathways.

**Figure 7 ijms-25-09544-f007:**
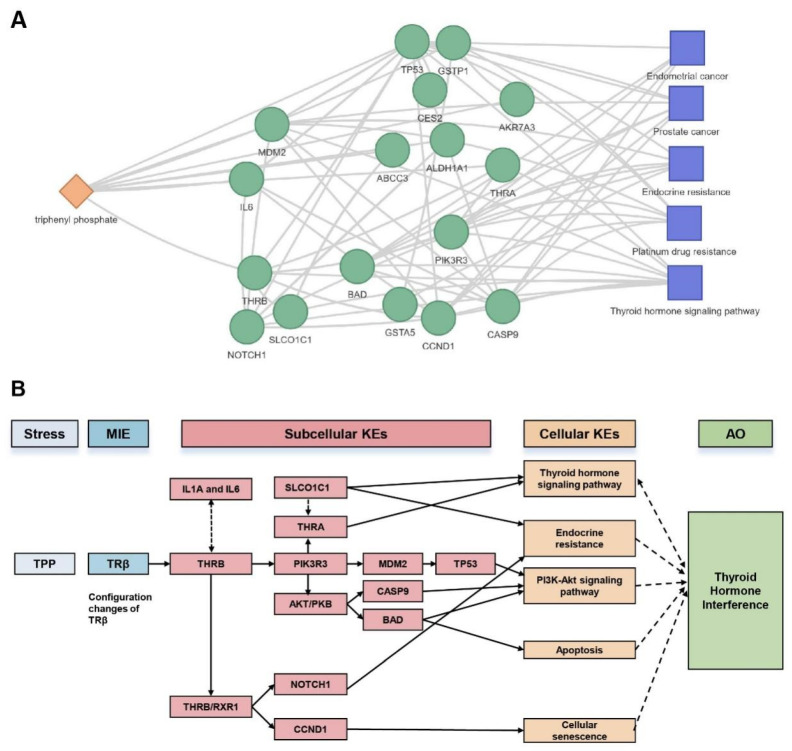
Knowledge Graph Analysis of Differential Genes in TPP Toxicity. (**A**) Knowledge Graph of TPP Toxicological Mechanism. In this network diagram, red nodes represent the toxin, yellow circular nodes represent genes, and green square nodes represent biological pathways. The analysis of key differential genes has revealed interactions between genes and the biological pathways in which they jointly participate. Utilizing the annotation library built into the software package, molecules directly interacting with TPP (marked with an asterisk, THRB) were identified, and biological pathways related to TPP were determined, ultimately presenting a network diagram associating the toxin, genes, and biological pathways. (**B**) AOP (Adverse Outcome Pathway) Network of TPP Toxicological Mechanism. In this diagram, “Stress” represents the exogenous toxin, “MIE” indicates the key molecule affected by the toxin, “Subcellular KEs” represents the key events occurring at the subcellular level, that is, a series of interactions between molecules triggered after the toxin affects the key molecule, “Cellular KEs” signifies the key events at the cellular level, that is, changes in biological pathways caused by molecular alterations, “AO” represents the adverse outcomes triggered by these changes. This AOP network intuitively presents the potential toxicological mechanism of TPP.

**Table 1 ijms-25-09544-t001:** Integrated External Software Packages.

External Software Package	Version	Functionality
NormExpression	V0.1.0	getNormMatrix; gatherCVs
ggord	V1.1.7	ggord.pca
limma	V3.54.2	model.matrix; lmFit; eBayes
clusterProfiler	V4.6.2	enricher
org.Hs.eg.db	V3.16.0	org.Hs.eg.db
gprofiler2	V0.2.1	gconvert
fgsea	V1.24.0	fgsea; plotEnrichment
igraph	V1.3.5	graph_from_edgelist; clusters;layout
msigdbr	V7.5.1	msigdbr
ComplexHeatmap	V2.14.0	rowAnnotation; Heatmap
dplyr	V1.0.10	mutate;select; group_by
ggplot2	V3.4.0	ggplot; ggtitle; theme; geom_point; geom_hline

**Table 2 ijms-25-09544-t002:** Data Sources.

Source	Version	URL
CTD	v2021-10	http://ctdbase.org/, accessed on 13 October 2021
AOP-Wiki	v2022-12	https://aopwiki.org/, accessed on 10 December 2022.
KEGG	v0.7.2	https://www.kegg.jp/, accessed on 5 October 2020.
DrugBank	v2020-12-15	https://go.drugbank.com/, accessed on 15 December 2020.
DisGeNet	v7.0	https://www.disgenet.org/, accessed on 15 October 2022.
Disease Ontology	v2021-10-11	https://disease-ontology.org/, accessed on 11 October 2022.
Human Phenotype Ontology	v2021-10-10	https://hpo.jax.org/app/, accessed on 10 October 2022.
PhosphoSitePlus	v6.6	https://www.phosphosite.org/
UbiBrowser	v2.0	http://ubibrowser.ncpsb.org.cn, accessed on 18 October 2022.
ENCODE	v120	https://www.encodeproject.org/, accessed on 8 October 2022.
STRINGdb	v11.0	https://string-db.org/, accessed on 18 December 2020.

## Data Availability

All the codes and datasets are packaged as ToxDAR and available at https://github.com/TMCjp/ToxDAR.

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
