# Peer review of "ToxDAR: A Workflow Software for Analyzing Toxicologically Relevant Proteomic and Transcriptomic Data, from Data Preparation to Toxicological Mechanism Elucidation"

_ijms, 2024, doi:10.3390/ijms25179544_

Round 1

Reviewer 1 Report

Comments and Suggestions for Authors

General comments to the authors.

The article entitled " ToxDAR: A Workflow Software for Analyzing Toxicologically Relevant Proteomic and Transcriptomic Data, From Data Preparation to Toxicological Mechanism Elucidation ", is well written and presents interesting data and analyzes in the medical chemistry expertise field. the authors developed ToxDAR, a workflow-oriented R package for preprocessing and analysis of multiomics toxicology data. However, it has some methodological and manuscript layout failures that strongly need to be considered by the authors before being accepted for publication. 

1- The introduction does not provide a consistent reference on other toxicity knowledge-based programs available for this category of study.

2 - What is the main objective of the ToxDAR software and how does it help in the analysis of toxicological data?

3 - What are the four main modules of ToxDAR and what functions does each one perform?

4 - The manuscript does not elucidate information such as dose-response effect prediction and other weaknesses that have not been overcome in this field of study and are widely recognized by those who study new alternatives for toxicity programs in a few days.

5 - What normalization methods are integrated into ToxDAR and how are they evaluated?

6 - How does ToxDAR perform functional enrichment analysis and what types of data does it use for this?

7 - There is no tutorial or illustration on how to use ToxDAR, which leaves readers without any alternative to evaluate the tool and reproduce information mentioned in the text of the article, reducing the credibility of the presented proposal. I strongly recommend that authors include a self-explanatory flowchart in the supplementary material section on how to use the program.

Author Response

Comments 1: The introduction does not provide a consistent reference on other toxicity knowledge-based programs available for this category of study.

Response 1: Thanks for your comment.

Currently, there have been some efforts in building toxicology-related knowledge bases. For example, the National Center for Toxicogenomics has developed the Chemical Effects in Biological Systems (CEBS) knowledge base (EHP Toxicogenomics. 111(1T):15-28.) which includes toxicology-related literature and dataset information. The Comparative Toxicogenomics Database (CTD)integrates toxicology information on chemicals, genes, phenotypes, and exposures, revealing the impact of environmental factors on disease etiology and molecular mechanisms (Nucleic Acids Research, 49: D1138). The ECOTOXicology Knowledgebase aggregates various ecological toxicity data and supports risk assessment for ecological toxicity testing (Environ Toxicol Chem. 41(6):1520-1539.). However, the field of toxicology still lacks a comprehensive knowledge graph that systematically presents the relationships between toxic molecules and biological systems, as well as omics data analysis software based on such a knowledge graph. The ToxDAR tool we provide has addressed this challenge.

We have made corresponding revisions in the introduction section.

Comments 2: What is the main objective of the ToxDAR software and how does it help in the analysis of toxicological data?

Response 2: Thanks for your comment.

Despite the substantial accumulation of toxicogenomics data, identifying relevant biological pathways associated with toxicity and linking omics data with existing toxicological knowledge remains a critical challenge. This is the main objective of the ToxDAR software developed in our article.

In the paper, we use the toxicogenomics dataset of triphenyl phosphate (TPP) as a case study to demonstrate the application of ToxDAR. The software reveals information such as differential genes and pathways reflected in the toxicogenomics data while indicating the correlation between toxicological data and existing knowledge about TPP. The specifics are as follows:

  • Data Preprocessing: We employed ten normalization methods integrated within ToxDAR to eliminate technical biases and batch effects. The normalization effectiveness was evaluated using AUCVC and mSCC metrics, with the Upper Quartile (UQ) method selected as the optimal normalization approach.
  • Quality Control: The data quality control module of ToxDAR generated PCA and violin plots, visually illustrating the variability and distribution of gene expression across different TPP concentration groups and within the same group. The results indicated high data quality, suitable for subsequent analysis.
  • Data Analysis: Differential expression analysis was used to identify genes associated with TPP exposure at different concentrations and time points. Volcano plots displayed upregulated and downregulated genes. Enrichment analyses using annotation datasets such as GO, DO, and KEGG were performed to generate GO functional enrichment plots, revealing associations between TPP exposure and signaling pathways related to cellular apoptosis and inflammatory response. Key molecules like ABCC3 and THRB were highlighted. ssGSEA analysis showed that at a TPP concentration of 881 mg/kg, the apoptosis pathway was significantly activated, while the phenylalanine metabolism pathway was significantly suppressed. Additionally, the trend of activation of inflammation-related pathways increased with TPP concentration.
  • Data Interpretation: Using the ToxDAR knowledge annotation module, we constructed a knowledge graph mapping the relationships between toxins, key molecules, biological pathways, and phenotypes. This revealed that TPP might impact thyroid function by activating thyroid hormone receptor β (THRB), affecting signaling pathways related to cellular apoptosis and inflammatory response.

A complete tutorial on the above analysis is available on GitHub (https://github.com/TMCjp/

ToxDAR).

We have made corresponding revisions in the “2. Result-2.1 Software framework” section.

Comments 3: What are the four main modules of ToxDAR and what functions does each one perform?

Response 3:  Thanks for your comment.

ToxDAR includes four main modules: Data Preparation, Quality Control, Data Analysis, and Data Interpretation.

(I) The first module, Data Preparation, is designed to address batch effects in toxicogenomics data. It integrates ten standardization techniques and evaluates the results using AUCVC and mSCC metrics.

(II) The second module, Quality Control, focuses on testing the reliability of toxicogenomics data and performing screening. It employs PCA techniques for dimensionality reduction and feature extraction, and uses violin plots to compare and assess different experimental groups.

(III) The third module, Data Analysis, consists of the Differential Analysis Module, Functional Annotation Module, and Network Analysis Module. The Differential Analysis Module primarily identifies differentially expressed genes based on the DESeq2 framework, and further analyzes toxin-related genes through intersection analysis and literature evidence. The Functional Annotation Module reveals the biological functions and pathway activation of relevant genes through enrichment analysis of differentially expressed genes and various graphical representations. The Network Analysis Module constructs and analyzes molecular interaction networks to uncover the relationships between toxins and key genes and biological processes.

(IV)The fourth module, Data Interpretation, integrates toxicological knowledge and provides information on five types of entities and fifteen types of entity relationships, offering comprehensive annotations of the analysis results.

We have made corresponding revisions in the “2. Result -2.2 Software function” section and Figure 1.

Comments 4: The manuscript does not elucidate information such as dose-response effect prediction and other weaknesses that have not been overcome in this field of study and are widely recognized by those who study new alternatives for toxicity programs in a few days.

Response 4:  Thanks for your great suggestion.

Our software can be applied to various issues in the field of toxicology research, such as dose-effect prediction and the exploration of new alternative for toxicity programs. In dose-effect prediction, our software can examine the impact of different doses of toxins on the organism and molecular changes to determine the effect of dosage on toxicological mechanisms. This provides possibilities and support for further establishing dose-response relationships in toxicology. In the case study of triphenyl phosphate (TPP) that we provide, we explore the changes in differentially expressed genes and the effects on biological pathway activities in the L02 cell line at various concentrations of TPP.

In terms of new alternative for toxicity programs, because traditional animal models may not accurately reflect the actual conditions of clinical patients and exhibit species differences, the academic community has gradually developed new systems such as in vitro cell models like organoids. These systems include liver, kidney, and lung organ-on-chip models and multi-organ-on-chip models designed for toxicological assessment (Pharmacological research, 169 (2021): 105608; Environment International, 184: 108415). Our software can evaluate the accuracy and stability of these organ-on-chip models by examining the differences in toxin-responsive protein expression and pathway activities between organ-on-chip models and traditional animal models.

We have made corresponding revisions in the discussion section.

Comments 5: What normalization methods are integrated into ToxDAR and how are they evaluated?

Response 5: Thanks for your comment.

     1.Ten normalization methods:

In our article, we integrated ten normalization methods: the median of the ratios of observed counts (DESeq), upper quartile (UQ), Trimmed Mean of M values (TMM), Total Ubiquitous (TU), Total Read Count (TC), Total Read Number (TN), External RNA Control Consortium (ERCC), Housekeeping Genes (HG7), Cellular RNA (CR), and Nuclear RNA (NR). These methods offer significant advantages in normalizing omics data.

  • The DESeq method is a differential expression analysis approach based on a negative binomial distribution model. Its underlying principle involves linking variance to mean through local regression, allowing for precise estimation of gene expression variability. The research team led by Simon Anders tested DESeq across various RNA-Seq datasets and found that it not only employs a more robust size estimation formula but also offers faster computation and a more straightforward conceptual framework (Genome Biol. 11(10): R106.). Juliana Costa-Silva's team evaluated several RNA-Seq differential expression analysis methods using qRT-PCR data and concluded that DESeq2 provides the most balanced performance in terms of precision, accuracy, and sensitivity (PLoS One. 12(12): e0190152.). The Upper Quartile (UQ) method is a statistical approach designed to reduce the impact of extreme values on estimation accuracy. It works by normalizing the raw expression levels of each gene by dividing them by the upper quartile value. Pierre R. Bushel's team evaluated seven different normalization methods on human HepaRG cell control samples and found that the Upper Quartile (UQ) method performed best in maintaining fold-change (FC) levels (Frontiers in Genetics, 11: 594). Bullard et al. further validated that the UQ method's results were consistent with those obtained using edgeR, with agreement up to 10 decimal places, demonstrating that UQ is a rigorously tested and reliable tool suitable for high-precision data normalization (BMC Bioinformatics. 2010; 11:94.).
  • The Trimmed Mean of M-values (TMM) method is a statistical approach designed to address the impact of sequencing depth differences between samples on gene expression comparisons. The principle behind TMM involves calculating log-ratios, removing extreme values, and computing the mean to normalize gene expression data. Robinson and colleagues successfully applied the TMM method to normalize samples with varying types and amounts of RNA, effectively reducing the artifacts caused by differences in RNA composition between samples (Genome Biol. 11(3): R25.). Yingdong Zhao and others found that, in downstream analyses of PDX RNA-seq data, the TMM method outperformed TPM and FPKM methods in cross-sample comparisons and differential expression analysis (J Transl Med. 2021;19(1):269.).
  • The Total Ubiquitous (TU) method is a normalization technique used in gene expression data analysis. It operates by identifying genes that are ubiquitously expressed across most samples and uses the expression levels of these genes to calculate a global normalization factor between samples.
  • The Total Read Count (TC) method is a simple and intuitive normalization approach where the raw read count for each gene is divided by the total read count (library size) of that sample, thereby reducing the impact of sequencing depth on gene expression levels.
  • The Total Read Number (TN) method follows the same principle as the Total Read Count (TC) method. However, in certain contexts, TN may refer to the total number of reads, while TC could specifically refer to the reads used for analysis. In studies involving the NormExpression package, normalization using the TU, TC, and TN methods has been demonstrated to be effective in both single-cell RNA-seq and bulk RNA-seq data (Frontiers in genetics,2019;10: 400.).
  • The External RNA Control Consortium (ERCC) method is an RNA sequencing normalization technique that uses external RNA control samples of known concentrations to correct for technical variations and differences in sequencing depth in gene expression data. Alison S. Devonshire and her team evaluated the utility of ERCC RNA standards in normalizing gene expression biomarker measurements. Their study found that ERCC RNA standards provide a universal approach to assess various aspects of platform performance and can offer technical variation information related to the quantification of biomarkers across different physiological abundance levels. The different combinations of these standards can serve as an ideal quality control toolkit for determining the accuracy of differences between normal and disease profiles (BMC Genomics. 2010;11: 662.).
  • The Housekeeping Genes (HG7) method is a gene expression data normalization technique that uses a set of stably expressed reference genes to correct for differences in gene expression between samples. In experiments such as quantitative PCR (qPCR) or Northern blot, the relative expression changes of the target gene can be assessed by comparing its expression level to that of the Housekeeping Genes (BMC genomics, 2007;8: 1.).
  • The Cellular RNA (CR) method is a normalization technique that uses the total RNA content within cells to correct for technical variations and differences in sequencing depth in gene expression data. Deeptiman Chatterjee and his team discussed the standardization of single-cell RNA sequencing workflows, which is a crucial step in studying cell-type-specific gene expression in tissues such as Drosophila ovaries. The paper highlights various methods for normalizing single-cell RNA sequencing data, including the CR method, which contribute to improving the accuracy and reliability of data analysis (Methods and Protocols 2023 Jul 19 (pp. 151-171.). New York, NY: Springer US.
  • The Nuclear RNA (NR) method is a normalization technique that uses the amount of RNA within the cell nucleus to correct for technical variations and differences in sequencing depth in gene expression data. Ding et al. conducted a systematic comparison of single-cell and single-nucleus RNA sequencing methods. Their study provides an in-depth analysis of various scRNA-seq approaches, demonstrating the practical application and importance of using the NR method for normalization. The research highlights the advantages of the NR method in improving data quality and analysis accuracy. This approach enables researchers to gain a deeper understanding of gene expression regulation mechanisms and their roles in different cell types or disease states (Nat Biotechnol. 38(6):737-746.). 

      2. Metrics to evaluates normalization results:

The ToxDAR platform evaluates normalization results using two metrics: AUCVC (Area Under the Curve of the Variation Coefficient Threshold Curve) and mSCC (Median Spearman Correlation Coefficient). The AUCVC metric represents the change in the number of uniform genes (defined as genes with normalized expression levels below a predefined threshold across all samples) as the threshold for the variation coefficient changes. A lower coefficient of variation (CV) for uniform genes indicates higher consistency between samples, suggesting that technical noise has been more effectively reduced. The mSCC metric reflects the proportion of gene pairs in the entire dataset that have corresponding Spearman correlation coefficients. By calculating the Spearman correlation coefficient between each pair of genes (normalized versus original data), we obtain the median Spearman correlation coefficient across all gene pairs to assess the effectiveness of the data normalization strategy. In the context of gene expression analysis, an mSCC value close to zero indicates a low correlation between the normalized data and the original data at the gene expression level.

We have made corresponding revisions in the “2. Result -2.2 Software function” section and added Supplementary Material S3.

Comments 6: How does ToxDAR perform functional enrichment analysis and what types of data does it use for this?

Response 6: Thanks for your comment.

      1.Details of functional enrichment analysis:

In this article, we investigate the most commonly used method for differential gene enrichment analysis—the hypergeometric distribution (Bioinformatics. 31(10):1592-8, which has been applied in hundreds of published studies. This method consists of three steps: identifying differentially expressed genes, annotating these genes with respect to their involvement in pathways and processes, and performing statistical tests to determine whether these genes are significantly enriched in biological processes.

In the first step, differentially expressed genes are identified using statistical models, and genes with a p-value < 0.05 are selected as significantly differentially expressed. The probability of gene expression differences is also estimated. In the second step, annotation is performed through Gene Ontology (GO) terms (Nat Genet. 25(1):25-9.) or Kyoto Encyclopedia of Genes and Genomes (KEGG) pathways (Nucleic Acids Res. 28(1):27-30.) to determine the roles of these genes in biological processes. In the final step, the hypergeometric distribution is used for functional enrichment analysis of differential genes. The hypergeometric distribution describes the statistical probability of the number of genes in a specific function or pathway relative to the total number of genes in the differential expression gene set, helping us determine the extent of significant enrichment in specific functions or pathways. Genes with a p-value < 0.05 are selected as significantly differentially expressed gene sets. Since functional enrichment analysis involves multiple categories, p-values for each category need to be adjusted for multiple comparisons to control the false discovery rate.

         2. Type of functional category data:

In our functional enrichment analysis, the underlying functional data used are GOCC, GOBP, GOMF (Nat Genet. 25(1):25-9.), and KEGG (Nucleic Acids Res. 28(1):27-30.): (1) Cellular Component (CC): This level describes the localization of gene products (e.g., proteins) within the cell. It helps to understand the function and pathways of gene products within the cell. (2) Biological Process (BP): This level describes the biological processes that the genes participate in. It helps to understand the physiological functions of the organism and the mechanisms of disease development. (3) Molecular Function (MF): This level describes the molecular-level functions of gene products, which helps in understanding their interactions with other molecules or in catalyzing biochemical reactions. (4) KEGG enrichment analysis is a database resource used for functional annotation and pathway enrichment analysis of a set of genes. By comparing the gene set with pathway annotations in the KEGG database, it identifies pathways that are significantly enriched in specific biological processes or diseases. This helps provide a deeper understanding of the functional impact of the gene set.

We have made corresponding revisions in the “Software section's functional Annotation Module” section and added Supplementary Material S3.

Comments 7: There is no tutorial or illustration on how to use ToxDAR, which leaves readers without any alternative to evaluate the tool and reproduce information mentioned in the text of the article, reducing the credibility of the presented proposal. I strongly recommend that authors include a self-explanatory flowchart in the supplementary material section on how to use the program.

Response 7:  Thank you for your great suggestions.

We have added the flowcharts to both GitHub (https://github.com/TMCjp/ToxDAR), providing detailed instructions on how to use the various modules and functions of ToxDAR. The provided script with code and annotation information will help readers evaluate the ToxDAR tool and reproduce the TPP transcriptomic data example mentioned in the text. Users can perform the example analysis by following the step-by-step instructions available in the GitHub and the appendix.

Reviewer 2 Report

Comments and Suggestions for Authors

The authors presented a pipeline for analyzing proteomic and transcriptomic data that are specially related to toxicology. It can be useful for the community. I don't have any comments on the manuscript since the pipeline is straightforward. However, I do think the GitHub page is not in publishable stage. I'd recommend the authors to polish their GitHub page, which will actually be visited by the users more than their manuscript if people found the pipeline useful. At minimal, they should have an introduction page, a manual page, and a vignette page. Nowadays, detailed documentation for software and pipeline is very important. 

Author Response

Comments 1: The authors presented a pipeline for analyzing proteomic and transcriptomic data that are specially related to toxicology. It can be useful for the community. l don't have any comments on the manuscript since the pipeline is straightforward. However, l do think the GitHub page is not in publishable stage. I'd recommend the authors to polish their GitHub page, which will actually be visited by the users more than their manuscript if people found the pipeline useful. At minimal, they should have an introduction page, a manual page, and a vignette page. Nowadays, detailed documentation for software and pipeline is very important.

Response 1: Thanks for your high evaluation on our manuscript.

We have updated the introduction page on GitHub as requested.

The content includes the following six sections: an overview of ToxDAR, the composition of functional modules, required software packages, code installation requirements, the main workflow of example analysis, and potential errors and their solutions.

For details, please refer to the link: https://github.com/TMCjp/ToxDAR.

Round 2

Reviewer 1 Report

Comments and Suggestions for Authors

The authors responded to all the comments made in the first evaluation of the manuscript. Therefore, I consider the manuscript appropriate for future acceptance and publication.

Reviewer 2 Report

Comments and Suggestions for Authors

It looks well.